# An Algorithm for Generating Outdoor Floor Plans and 3D Models of Rural Houses Based on Backpack LiDAR

**DOI:** 10.3390/s24175723

**Published:** 2024-09-03

**Authors:** Quanshun Zhu, Bingjie Zhang, Lailiang Cai

**Affiliations:** 1School of Surveying and Land Information Engineering, Henan Polytechnic University, Jiaozuo 454003, China; zhuquanshun@home.hpu.edu.cn; 2Shanghai Huace Navigation Technology Ltd., Shanghai 201700, China; zhangbingjie@home.hpu.edu.cn

**Keywords:** backpack LiDAR, outdoor floor plan, three-dimensional model, digital countryside

## Abstract

As the Rural Revitalization Strategy continues to progress, there is an increasing demand for the digitization of rural houses, roads, and roadside trees. Given the characteristics of rural areas, such as narrow roads, high building density, and low-rise buildings, the precise and automated generation of outdoor floor plans and 3D models for rural areas is the core research issue of this paper. The specific research content is as follows: Using the point cloud data of the outer walls of rural houses collected by backpack LiDAR as the data source, this paper proposes an algorithm for drawing outdoor floor plans based on the topological relationship of sliced and rasterized wall point clouds. This algorithm aims to meet the needs of periodically updating large-scale rural house floor plans. By comparing the coordinates of house corner points measured with RTK, it is verified that the floor plans drawn by this algorithm can meet the accuracy requirements of 1:1000 topographic maps. Additionally, based on the generated outdoor floor plans, this paper proposes an algorithm for quickly generating outdoor 3D models of rural houses using the height information of wall point clouds. This algorithm can quickly generate outdoor 3D models of rural houses by longitudinally stretching the floor plans, meeting the requirements for 3D models in spatial analyses such as lighting and inundation. By measuring the distance from the wall point clouds to the 3D models and conducting statistical analysis, results show that the distances are concentrated between −0.1 m and 0.1 m. The 3D model generated by the method proposed in this paper can be used as one of the basic data for real 3D construction.

## 1. Introduction

The traditional steps for generating indoor and outdoor floor plans of buildings typically include several stages: First, instruments such as total stations and RTK are used to collect the coordinates of the corners of the house walls, or tools like rangefinders and tape measures are employed to measure the dimensions of walls, doors, and windows. Then, the data is manually drawn based on field sketches. This method is not only labor-intensive but also prone to data omissions. In recent years, with the development of LiDAR technology [1,2,3], new methods have gradually been proposed. These methods first use LiDAR to collect point cloud data of the building’s interior and exterior, and then analyze and process the point cloud data to ultimately achieve the automated generation of indoor and outdoor floor plans. These methods have become a research hotspot [4,5,6,7]. Many scholars have conducted in-depth studies on this issue and proposed various interrelated yet distinct methods.

Based on relevant literature, the steps for generating building floor plans using geometric and mathematical methods are roughly as follows [8,9,10,11,12,13]: first, extract the point cloud data of the building walls, then project these wall point clouds onto a horizontal plane, and finally, draw the floor plan from the projected point clouds. The methods for extracting wall point clouds involve directly extracting point clouds from the middle of the building. This is done by selecting an appropriate height range based on the building structure and extracting the point cloud data from the middle, as shown in Figure 1a; Statistical Histogram Based on Building Elevation Information. Figure 1b shows a histogram of building elevation information generated from indoor point cloud data collected by backpack LiDAR. The indoor data includes point clouds of the ground and roof, better reflecting the uniformity of point cloud distribution by using the Random Sample Consensus (RANSAC) algorithm to extract wall point clouds, and identifying and extract wall point clouds through the RANSAC algorithm. For the extracted wall point clouds, the methods for drawing floor plans can be roughly categorized as follows [14,15]: Direct Projection and Resampling Method: Directly project the wall point clouds onto a horizontal plane and resample them, using the vector angles and distances of the point clouds to complete the floor plan drawing; Depth Image Conversion Method: Convert the wall point clouds into a depth image on the horizontal plane and analyze each line in the depth image to detect the actual wall positions; Polygon Partitioning and Energy Minimization Method: Project the wall point clouds onto a horizontal plane to generate multiple polygon partitions, and use an energy minimization algorithm to select the boundaries that belong to the walls, thus completing the floor plan drawing. In recent years, using deep learning to generate building floor plans has become a new research focus [16,17]. For example, the Scan2Plan model uses deep neural networks to cluster unordered point clouds in the initial stage, then predicts the corner points of each room and draw the floor plans, finally stitching together the floor plans of each room for output. Another model, FloorPP-Net, converts building point clouds into point pillars, predicts the corner points and wall positions, and thus generates the floor plans.

Building on the generation of outdoor floor plans, further research can be conducted to extend these into 3D models of buildings [18,19]. Current research focuses on using LiDAR to collect 3D coordinate data and image data of building surfaces. By analyzing this data, 3D models of buildings are generated, and image data is used for coloring, as shown in Figure 2. Many scholars have conducted in-depth research on this issue and proposed various methods. These methods can be broadly classified into automated drawing methods based on spatial geometry, analytic geometry, and deep learning, as well as manual drawing methods using existing 3D modeling software (such as Revit, ArchiCAD or Rhino).

The methods for generating 3D models of buildings based on spatial geometry can be broadly classified into three categories [20,21,22]: Directly Generating 3D Models from Raw Point Cloud Data: For example, using the Delaunay triangulation method to generate 3D models from point clouds of mine tunnels; Based on Feature Points, Feature Lines, or Wireframe Models of Building Walls: Using these features and their topological relationships to draw 3D models; Semantic Clustering of Building Point Clouds: Extracting objects such as walls, doors, and windows, drawing their 3D models separately, and then merging them. The methods for generating 3D models of buildings based on analytic geometry can also be roughly classified into three categories [23,24,25]: Poisson Surface Reconstruction Method: Effectively removing noise points from the point cloud; Non-Uniform Rational B-Splines (NURBS) Method: Constraining point cloud data to complete the 3D model drawing; Partial Differential Equation (PDE) Method: Dividing the point cloud into multiple surface segments expressed by PDEs, then merging these segments to reconstruct the 3D structure of complex surfaces. With advancements in computer hardware capabilities, deep learning-based methods for generating 3D models of buildings have also increased [26,27,28]. For example, Points2Surf Model reconstructs 3D models of objects directly from raw point clouds without normal vector information, using detailed local patches and coarsely estimated global information. Using Point Net++, this model extracts the depth features of each point and cluster them, further determining the precise positions of building roof corner points, thereby generating 3D models from airborne LiDAR data. Currently, the most commonly used method for generating 3D models of buildings involves first using LiDAR equipment to collect point cloud data of the buildings, then importing this data into existing 3D modeling software (such as Revit, ArchiCAD or Rhino), and finally manually creating the models [29,30,31]. This method has been widely applied in fields such as heritage preservation, civil defense engineering, disaster assessment, and substations.

Through an in-depth analysis of current domestic and international research, it is evident that there is limited exploration of automated outdoor floor plan generation, particularly incorporating attributes such as building materials and floor plan area. Rural houses, which are generally low-rise and varied in appearance, often face interference from vehicles, trees, and other obstructions during LiDAR data collection. This increases the difficulty of using point cloud data for the automated generation of outdoor floor plans of rural houses. Additionally, the surveying industry requires periodic updates of large-scale, high-resolution outdoor building floor plans. Therefore, we propose an algorithm that projects the point clouds of rural house exterior walls onto a horizontal plane and draws the outdoor floor plans based on the topological relationships of the wall point clouds. Compared with RTK-measured house corner coordinates, the floor plans generated using backpack LiDAR-collected outdoor point clouds of rural houses meet the accuracy requirements of 1:1000 topographic maps. This algorithm can also add information such as the area, building materials, and number of floors of rural houses to the floor plans. In terms of 3D model generation, traditional spatial and analytic geometry methods are only suitable for small indoor spaces or small objects and are not appropriate for the automated generation of large-scale outdoor 3D models of rural houses. Deep learning methods have high computational hardware requirements and lack sufficient outdoor point cloud datasets of rural houses. Existing software-based drawing methods are suitable for detailed modeling of single buildings but are not conducive to the rapid generation of large-scale simple 3D models. Therefore, we propose an algorithm for the rapid generation of outdoor 3D models based on the outdoor floor plans of rural houses. To meet the needs of 3D models for masking analysis and lighting analysis in rural settings, we extended the algorithm for drawing outdoor floor plans of rural houses. This algorithm uses the height of the walls in the point cloud data to longitudinally stretch the floor plans, quickly generating outdoor 3D models of rural houses.

## 2. Algorithms for Drawing Outdoor Floor Plans and 3D Models

### 2.1. Principle of the Outdoor Floor Plan Drawing Algorithm

This paper proposes an algorithm for generating outdoor floor plans based on the topological relationships of wall slices gridded point clouds, using point clouds of rural residential exterior walls collected by backpack LiDAR as the data source. This method is designed to meet the demand for periodically updating large-scale rural residential floor plans. The algorithm first segments the exterior wall point clouds by elevation and projects the segmented wall point clouds horizontally, followed by gridding. Subsequently, the outdoor floor plan is generated based on the topological relationships between wall points and the threshold angles of line segments. Finally, the algorithm calculates the area and the number of floors of the residence. Figure 3 illustrates the flowchart of this algorithm.

Preprocessing of outdoor point clouds for rural houses:

The outdoor point cloud data of rural residences collected by backpack LiDAR typically include ground, vehicles, and trees. To better generate outdoor floor plans of rural residences (Figure 4a,b), it is necessary to first classify various point clouds using existing point cloud processing software.

2.Clipping outdoor wall point clouds based on elevation:

When collecting point clouds of residential walls along rural roads using backpack LiDAR, the process is often obstructed by parked cars, planted fruit trees, and placed trash bins, leading to holes in the wall point clouds. Therefore, to create an outdoor floor plan of rural residences (Figure 4c), wall point cloud data within the height range of 1.5 to 1.7 m above the ground should be extracted.

3.Rasterized horizontal projection of clipped wall point cloud:

Since the generated outdoor floor plan is independent of elevation, the segmented wall point clouds can be projected onto the horizontal plane. The number of grid cells in the X and Y directions for a specified grid cell size can be calculated using Equation (1), and each point can then be assigned to the corresponding grid cell using Equation (2). Finally, the centroid of all points within each grid cell can be calculated using Equation (3) to eliminate potentially redundant data (Figure 5).
(1)$$\left\{\begin{array}{c} C_{X}=\left[\frac{X_{\max}-X_{\min}}{S_{\mathrm{grid}}}\right]+1\\ C_{Y}=\left[\frac{Y_{\max}-Y_{\min}}{S_{\mathrm{grid}}}\right]+1 \end{array}\right.$$

In the formula: $X_{\max}$, $X_{\min}$, $Y_{\max}$, $Y_{\min}$ represent the maximum and minimum values of the *X* and *Y* coordinates in the point cloud; $S_{\mathrm{grid}}$ represents the size of each grid cell; $\left[*\right]$ denotes the floor function; $C_{X}$ and $C_{Y}$ represent the number of grid cells in the *X* and *Y* directions.
(2)$$\left\{\begin{array}{c} I_{x}=\left[\frac{X_{i}-X_{\min}}{S_{\mathrm{grid}}}\right]\\ I_{y}=\left[\frac{Y_{i}-Y_{\min}}{S_{\mathrm{grid}}}\right] \end{array}\right.$$

In the formula: $X_{i}$ and $Y_{i}$ represent the coordinates of the i-th point; $I_{x}$ and $I_{y}$ represent the row and column indices of the grid cell where the i-th point is located.
(3)$$\left\{\begin{array}{c} \bar{X}=\frac{1}{n}\sum_{i=1}^{n}X_{i}\\ \bar{Y}=\frac{1}{n}\sum_{i=1}^{n}Y_{i} \end{array}\right.$$

In the formula: $\bar{X}$, $\bar{Y}$ represents the centroid coordinates of all points in the current grid cell.

4.Drawing the floor plan using the topological relationships of wall points:

Due to the lack of topological relationships among the points in the horizontally projected wall point clouds, the following method can be used to restore these relationships for floor plan generation. First, select any point P from the gridded wall point cloud, and perform a K-nearest neighbor search centered on P (sorted in ascending order of distance from P), recording the closest point Q. Next, perform another K-nearest neighbor search centered on Q (sorted in ascending order of distance from Q). If the closest point M to Q has already been recorded, find the nearest unrecorded point N among Q’s K-nearest neighbors. This process is repeated until point P is recorded again, thus restoring the topological relationships among the wall points (Figure 6).

5.Simplify the floor plan using line segment angle threshold:

The floor plan generated from the gridded wall points often contains numerous redundant vertices at straight-line segments. To simplify the floor plan, the following steps can be taken: Starting from any endpoint in the original floor plan, check the angles between each pair of adjacent line segments. If the angle is close to 180°, remove the common point of the two adjacent line segments and connect them into a new line segment. The condition for stopping the simplification is by generating new line segments clockwise or counterclockwise according to the topological relationship of the initial plan vertices until the initial point is recorded again, thus simplifying the redundant values between wall points (Figure 7).

6.Calculate the area of the outdoor floor plan:

In mathematics, for any polygon in a Cartesian coordinate system, if the coordinates of each vertex $P_{1}\left(x_{1},y_{1}\right)$, $P_{2}\left(x_{2},y_{2}\right)$, ⋯, $P_{n}\left(x_{n},y_{n}\right)$ are known, and the vertices are arranged in a clockwise or counter-clockwise order, the area of the polygon can be calculated using the Shoelace theorem (Equation (4)). By substituting the vertex coordinates of the simplified floor plan into Equation (4) in sequence, the area of the floor plan can be obtained.
(4)$$A=\frac{1}{2}\left|\sum_{i=1}^{n}\left(x_{i}y_{i+1}-x_{i+1}y_{i}\right)\right|$$

In the formula: $\left|*\right|$ represents the absolute value; $x_{n+1}$ equals $x_{1}$; $y_{n+1}$ equals $y_{1}$.

7.Calculate the number of floors in the rural house:

From the pre-segmented rural residential wall point cloud, the maximum elevation $Z_{\max}$ and minimum elevation $Z_{\min}$ of the walls can be obtained. Given the specified height of each floor in the rural residence, the number of floors can be calculated using Equation (5).
(5)$$H_{h}=\left[\frac{Z_{\max}-Z_{\min}}{h}\right]$$

In the formula: $H_{h}$ represents the number of floors in the rural house; the height of each floor of the specified rural residence is denoted by “*h*”, with [ ] representing the floor function (rounding down).

8.Add attribute information to the outdoor floor plan;

In mathematics, for any polygon in a Cartesian coordinate system, if the coordinates of each vertex $P_{1}\left(x_{1},y_{1}\right)$, $P_{2}\left(x_{2},y_{2}\right)$, ⋯, $P_{n}\left(x_{n},y_{n}\right)$ are known, and the vertices are arranged in a clockwise or counter-clockwise order, the centroid coordinates of the polygon can be calculated using Equation (6). By substituting the vertex coordinates of the simplified floor plan into Equation (6) in sequence, the centroid position of the floor plan can be determined. The area of the floor plan, along with information about the building material structure and number of floors, can be added to the centroid position (Figure 8). Material information data is recorded during on-site collection and manually imported as attribute information before running the algorithm, after which it is automatically loaded into the floor plan.
(6)$$\left\{\begin{array}{c} x=\frac{1}{6A}\sum_{i=1}^{n}\left(x_{i}+x_{i+1}\right)\left(x_{i}y_{i+1}-x_{i+1}y_{i}\right)\\ y=\frac{1}{6A}\sum_{i=1}^{n}\left(y_{i}+y_{i+1}\right)\left(x_{i}y_{i+1}-x_{i+1}y_{i}\right) \end{array}\right.$$

In the formula: A represents the area of the polygon; $x_{n+1}$ equals $x_{1}$; $y_{n+1}$ equals $y_{1}$.

9.Add elevation annotations to the floor plan:

In step (1) of the point cloud preprocessing, the ground points classified can be used to add several elevation annotation points around the outdoor floor plan (Figure 9). By randomly selecting a point P from the outdoor ground points collected by the backpack LiDAR, a radius search can be performed using P as the center and R as the radius to find its neighboring points. Using the coordinates of point P and its neighboring points, the plane position and elevation annotation points can be fitted using Equation (7).
(7)$$\left\{\begin{array}{c} \bar{x}=\frac{1}{k}\sum_{i}^{k}x_{i}\\ \bar{y}=\frac{1}{k}\sum_{i}^{k}y_{i}\\ \bar{z}=\frac{1}{k}\sum_{i}^{k}z_{i} \end{array}\right.$$

### 2.2. Principle of the Outdoor 3D Model Drawing Algorithm

This paper proposes an algorithm for quickly generating a 3D model of rural residential exteriors based on wall point cloud heights and outdoor floor plans, using point clouds of rural residential exterior walls collected by backpack LiDAR as the data source. This algorithm is designed to meet the needs of spatial analyses such as illumination and inundation. The algorithm first uses the exterior wall point clouds collected by backpack LiDAR to draw a floor plan. Then, it creates control points for the 3D model based on the wall heights and the horizontal positions of vertices in the floor plan. Finally, the exterior 3D model’s walls and roof are drawn sequentially according to the topological relationships of the vertices in the floor plan. Figure 9 shows the workflow of the algorithm for drawing the 3D model of a rural residence.

Drawing the outdoor floor plan of rural houses:

Since the 3D model of the rural residence exterior is based on the outdoor floor plan, the specific principles will not be described in detail here; please refer to Section 2.1 for more information.

2.Creating control points for the 3D model of rural houses:

From the rural residential exterior wall point clouds, the maximum and minimum wall heights $h_{\max}$ and $h_{\min}$ can easily be obtained. Then, the planar X and Y coordinates of each vertex in the floor plan are taken, and the maximum and minimum wall heights are assigned as the elevations of these vertices, respectively, to create control points for the 3D model of the rural residence’s exterior (Figure 10).

3.Drawing the walls of the outdoor 3D model of rural houses:

To draw the exterior 3D model’s walls according to the sequence of vertices in the floor plan, follow these steps. Step 1: Select any bottom point P as the starting point. Connect point P with its corresponding top point Q, then connect point Q with the adjacent bottom point M, and finally connect the bottom point M back to point P, completing the first triangular face. Step 2: Use point M as the starting point. Connect point M with its corresponding top point N, then connect point N with the adjacent top point Q, and finally connect the top point Q back to point M, completing the second triangular face. Repeat this process to complete all triangular faces (Figure 11).

4.Drawing the roof of the outdoor 3D model of rural houses:

To draw the exterior 3D model’s roof according to the sequence of vertices in the floor plan, follow these steps. Step 1: Select any top point P as the starting point for all triangular faces on the roof. Connect point P with its adjacent top point Q, then connect point Q with its adjacent top point M, and finally connect the top point M back to point P, completing the first triangular face. Step 2: Connect point P with point M, then connect point M with its adjacent top point N, and finally connect the top point N back to point P, completing the second triangular face. Repeat this process to complete all triangular faces (Figure 12).

5.Output the 3D model:

The three-dimensional model was exported in PLY file format. Figure 13 shows the outdoor 3D model of the rural residential area.

## 3. Data Collection

### 3.1. Use of Instruments

The backpack LiDAR (GEO-VISIOM, Beijing, China) used in this study is the SSW-QS lightweight indoor and outdoor measurement system with LiDAR SLAM technology (Figure 14). This instrument includes a panoramic camera, RTK, dual LiDAR, and a control system [32,33,34]. After LiDAR point cloud processing and resampling, the point cloud density is 1–3 cm. The instrument’s absolute accuracy is ≤10 cm, and relative accuracy is ≤3 cm. The point cloud has geographic references, using RTK-SLAM technology, with the coordinate system being CGCS2000, 3° zone.

The parameters of backpack laser SLAM system are shown in Table 1.

### 3.2. Study Area

This study focuses on a village near the suburbs (Figure 15). Through field surveys and satellite imagery, it was found that the village covers an area of about 100,000 square meters. The houses are arranged in two rows: one facing north and the other facing south. The residential buildings are typical self-built houses characteristic of rural northern China’s housing.

### 3.3. Operational Procedure

Based on the distribution of the village houses, the data collection strategy involved segmenting the outdoor roads and collecting point cloud data. To ensure data completeness and accuracy, the collection route was designed to form closed loops whenever possible and maintain symmetrical scanning in complex scenes. Starting from the center of the survey area, an 8-shaped route was adopted for circular closed-loop scanning. Figure 16 shows the walking route during data collection, automatically generated using the backpack LiDAR. Figure 17 shows the outdoor point cloud data of rural houses collected using the backpack LiDAR.

## 4. Accuracy Analysis

### 4.1. Parameter Optimization of the Outdoor Floor Plan Drawing Algorithm

In the outdoor floor plan drawing algorithm for rural houses designed in Section 2.1, parameters such as the height range for wall point cloud extraction, the grid size for wall point cloud rasterization, and the angle threshold for floor plan simplification are considered. Meanwhile, the 3D model algorithm for rural houses outlined in Section 2.2, as an extension of the floor plan drawing algorithm, does not require separate optimization of key parameters. Subsequently, this study will optimize the parameters needed for the outdoor floor plan drawing algorithm to ensure the best possible results in terms of the appearance and accuracy of the generated plans.

Height range for clipping wall point clouds:

To minimize point cloud voids caused by obstructions like vehicles, garbage bins, and air conditioning units, we selected four different height ranges for extraction: 1.3~1.5 m, 1.5~1.7 m, 1.7~1.9 m, and 1.9~2.1 m (Figure 18). Within the 1.3–1.5 m range, the walls are partially obscured by objects such as vehicles. Similarly, in the 1.9~2.1 m range, walls are affected by obstructions like air conditioning units. In contrast, the ranges of 1.5~1.7 m and 1.7~1.9 m show no such interference. Additionally, the wall uniformity in the 1.5~1.7 m range is superior to that in the 1.7~1.9 m range.

2.Grid size for rasterizing wall point clouds:

To minimize redundant point clouds after horizontal projection, we selected four different grid sizes: 0.05 m, 0.10 m, 0.20 m, and 0.40 m (Figure 19, where black points represent the projected wall points, and red points indicate the rasterized wall points). At grid sizes of 0.05 m and 0.10 m, redundant points appear in some wall areas, which affects the subsequent operation of connecting the initial floor plan. At grid sizes of 0.20 m and 0.40 m, this issue does not occur; however, the precision of the wall at 0.40 m is slightly lower than at 0.20 m.

3.Angle threshold for simplifying floor plans:

To reduce the number of vertices in the floor plan, we selected three different angle thresholds: 179.5°, 179.0°, and 178.0° (Figure 20, where black points represent the projected wall points, blue lines represent the floor plan connections, and red points represent the vertices of the floor plan). Analysis of the figure shows that with angle thresholds of 179.5° and 179.0°, the simplified floor plan closely matches the walls with no significant deviation. However, at an angle threshold of 178.0°, slight deviations appear in some local areas. Combined with the statistical results in Table 2, an angle threshold of 179.5° yields the best simplification effect, significantly reducing the number of vertices while minimizing the area loss of the floor plan.

### 4.2. Accuracy Analysis of Outdoor Floor Plans and 3D Models

Using the optimal parameters, the outdoor floor plan of rural houses was generated quickly (Figure 21), along with a simplified 3D model (Figure 22).

The horizontal positional accuracy of the floor plan and the simplified 3D model was compared and analyzed using RTK measurements of the coordinates of the corners of rural house walls (Figure 19 and Figure 23). According to the comparative analysis results in Table 3, the maximum distance between the floor plan wall corners and the RTK-measured coordinates on the horizontal plane was 8.6 cm, the minimum distance was 1.5 cm, the average distance was 4.9 cm, and the standard error was 1.9 cm. The outlier value calculated from the standard error and average distance was 0.106 m, and no observational deviations exceeding this value were found in the detailed check in Table 3. According to relevant standards, the floor plan generated from the outdoor point cloud of rural houses collected by the backpack-mounted LiDAR meets the planar accuracy requirements of a 1:1000 topographic map (Table 4).

Since the simplified 3D model is based on the floor plan, its horizontal positional accuracy is similar. As shown in Figure 24, the distances from the point cloud of the rural house exterior walls to the simplified 3D model are primarily concentrated between −0.1 m and 0.1 m. It can be concluded that the simplified 3D model created using the backpack-mounted LiDAR system in this study has high accuracy.

## 5. Conclusions

This paper investigates methods for the automated generation of outdoor floor plans and 3D models of rural houses using backpack LiDAR point clouds as the primary data source. The main conclusions are as follows:Automated generation of outdoor floor plans: Based on the point cloud data collected by backpack LiDAR, this paper proposes an outdoor plan drawing algorithm based on the topological relationship between slice and grid wall point clouds. By comparing with the house corner coordinates measured by RTK, it is verified that the plan drawn by this algorithm can meet the accuracy requirements of 1:1000 topographic map.Rapid automated generation of large-scale outdoor 3D models: To address the issue of quickly and automatically generating large-scale outdoor 3D models of rural houses, we proposed an algorithm for rapidly constructing 3D models based on outdoor floor plans. By measuring the distance from the wall point clouds to the 3D model and conducting statistical analysis, the results show that the distance is within 0.1 m.Optimization of key parameters in floor plan generation: The algorithm’s parameters, including the height range of the wall point clouds, the grid size for rasterizing the wall point clouds, and the angle threshold for simplifying the floor plan have been optimized. This ensures that the generated outdoor floor plans and 3D models of rural houses achieve the highest level of accuracy.The outdoor data of rural residences collected using backpack LiDAR in this study does not include roof data. For the roof, we plan to use drone LiDAR in the future to generate roof data, which will then be stitched together to make the 3D model more accurate. We also plan to use a total station to collect data in the test area during the next phase and compare it with the data collected using the backpack and RTK systems.

## Figures and Tables

**Figure 1 sensors-24-05723-f001:**
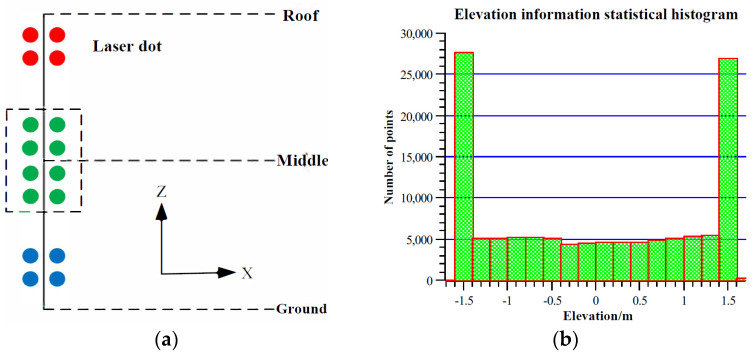
Method for extracting building wall point clouds. (**a**) Vertical segmentation of building point cloud (**b**) Histogram of indoor building elevation data.

**Figure 2 sensors-24-05723-f002:**
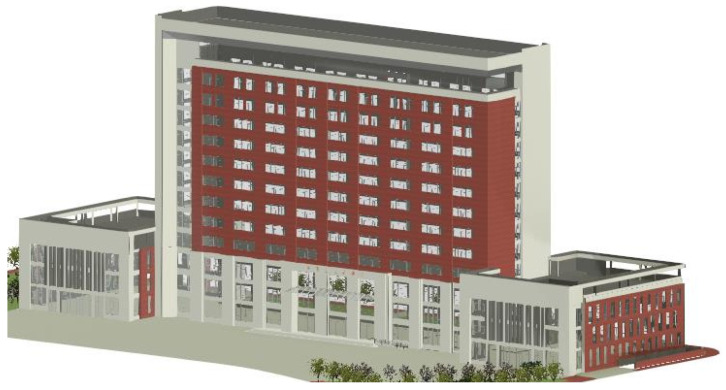
3D model of building exterior.

**Figure 3 sensors-24-05723-f003:**
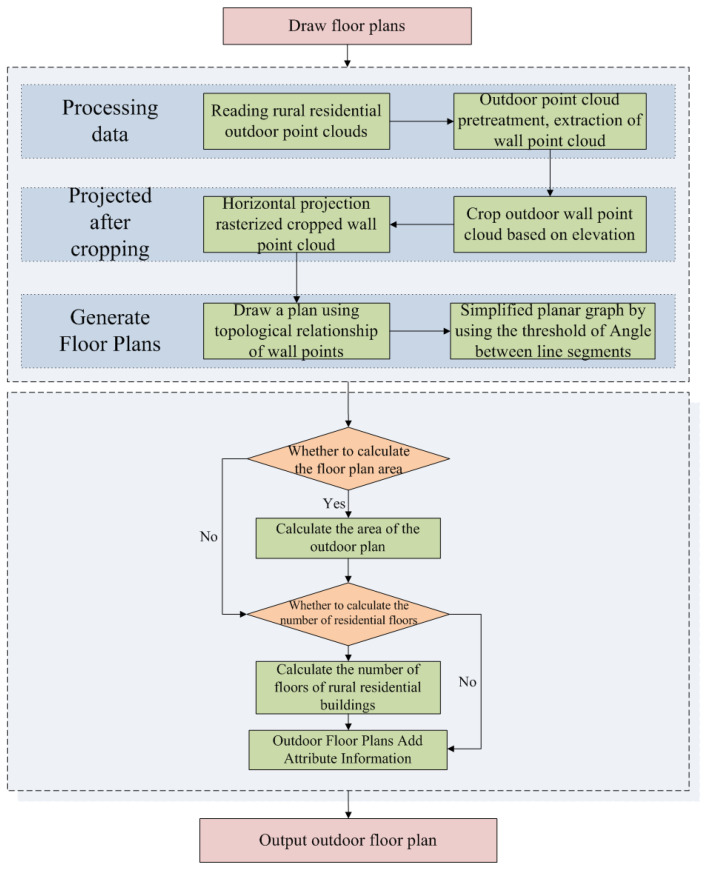
Flowchart of the algorithm for drawing outdoor floor plans of rural houses.

**Figure 4 sensors-24-05723-f004:**
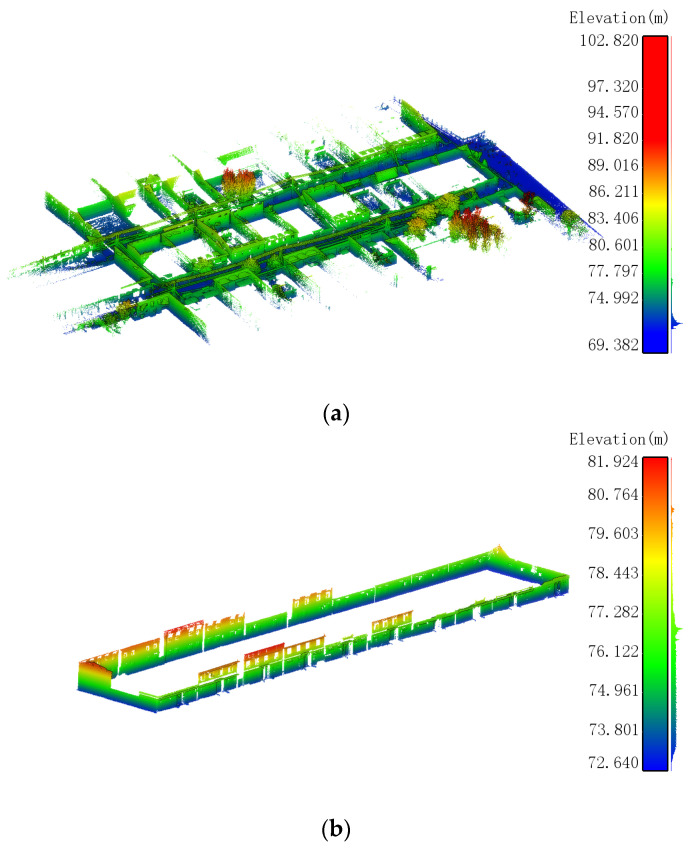
Preprocessing and clipping of outdoor point clouds for rural houses. (**a**) Outdoor point cloud of rural houses (**b**) Point cloud of exterior walls of rural houses (**c**) Clipped point cloud of exterior walls of rural houses.

**Figure 5 sensors-24-05723-f005:**
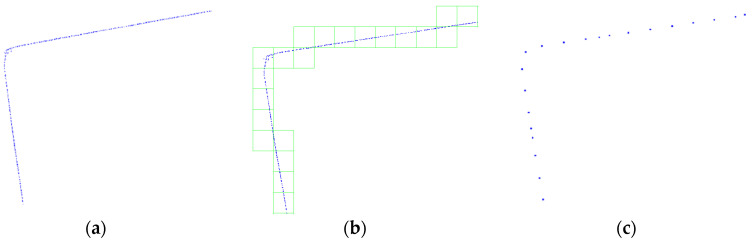
Diagram of rasterized horizontal projection of wall point cloud. (**a**) Point cloud after horizontal projection (wall corners) (**b**) Grid division (**c**) Calculating the centroid for each grid cell.

**Figure 6 sensors-24-05723-f006:**
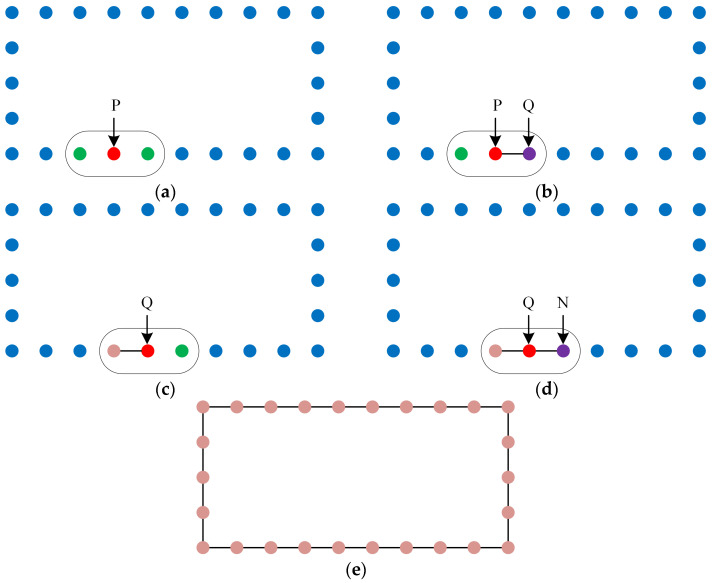
Diagram of drawing the floor plan using topological relationships of wall points. (**a**) Perform KNN search with point P as the center (**b**) Draw a line between point P and the nearest point Q (**c**) Perform KNN search with point Q as the center (**d**) Draw a line between point P and the nearest unconnected point N (**e**) Continue this process to draw the initial floor plan.

**Figure 7 sensors-24-05723-f007:**
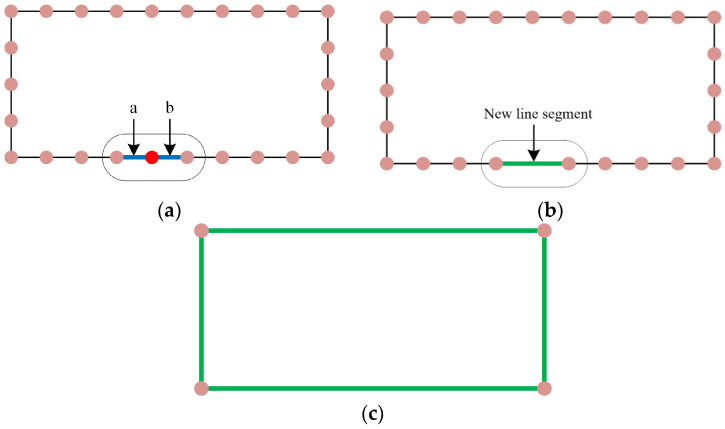
Diagram of simplifying the floor plan using line segment angle threshold. (**a**) Adjacent line segments a and b have an angle close to 180° (**b**) Remove the shared point of adjacent line segments a and b (**c**) Continue this process to simplify the floor plan.

**Figure 8 sensors-24-05723-f008:**
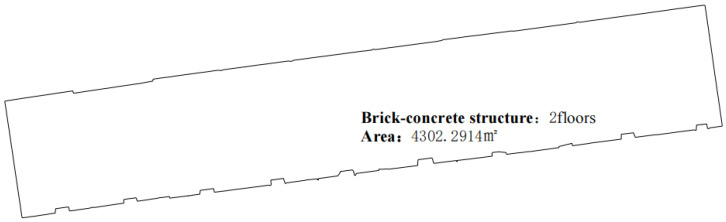
Outdoor floor plan of a rural house.

**Figure 9 sensors-24-05723-f009:**
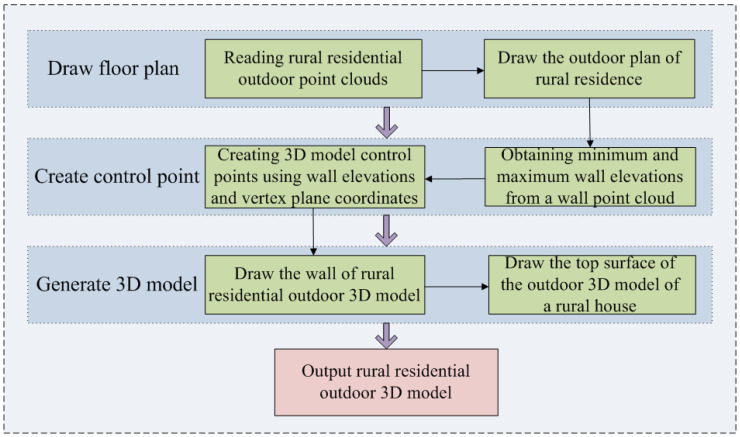
Flowchart of the algorithm for drawing 3D models of rural houses.

**Figure 10 sensors-24-05723-f010:**
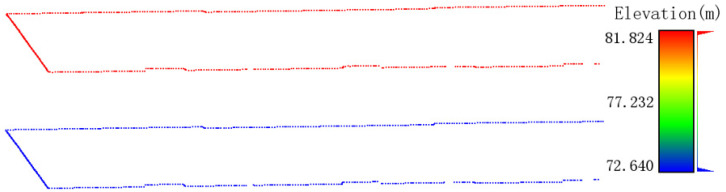
Control points of the outdoor 3D model for rural houses.

**Figure 11 sensors-24-05723-f011:**
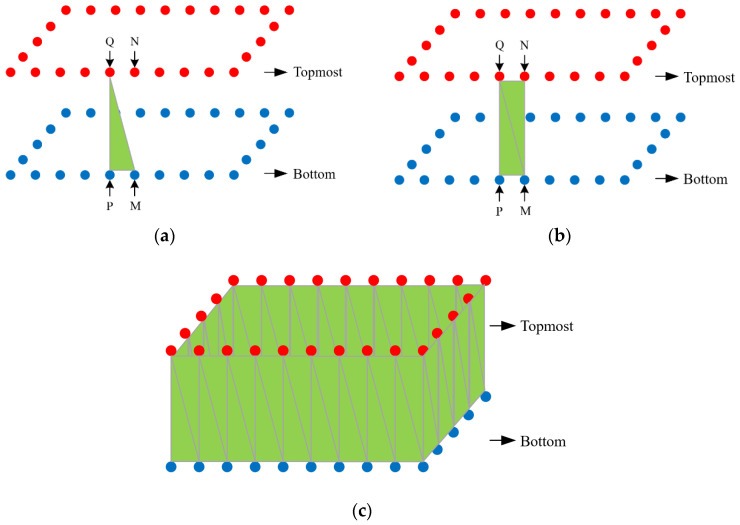
Diagram of drawing the walls of the outdoor 3D model of rural houses. (**a**) Connecting the first triangle (**b**) Connecting the second triangle (**c**) Continue this process to draw the walls of the outdoor 3D model.

**Figure 12 sensors-24-05723-f012:**
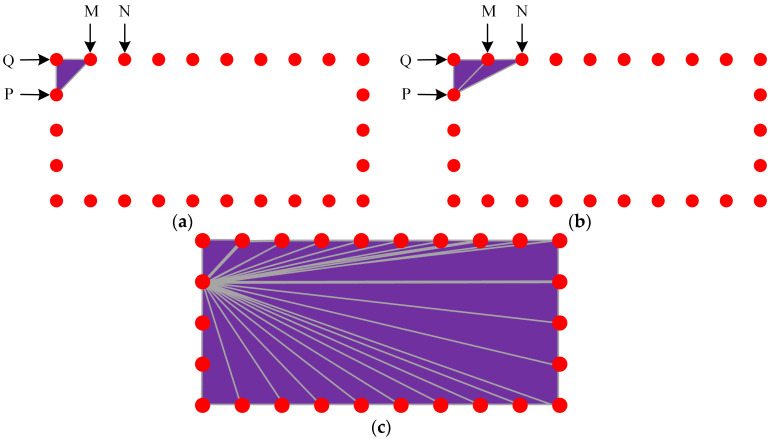
Diagram of drawing the roof of the outdoor 3D model of rural houses. (**a**) Connecting the first triangle (**b**) Connecting the second triangle (**c**) Continue this process to draw the roof of the outdoor 3D model.

**Figure 13 sensors-24-05723-f013:**
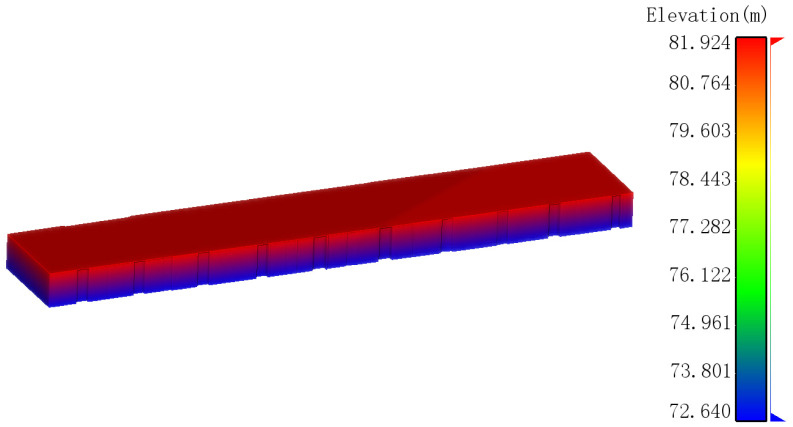
Outdoor 3D model of rural houses.

**Figure 14 sensors-24-05723-f014:**
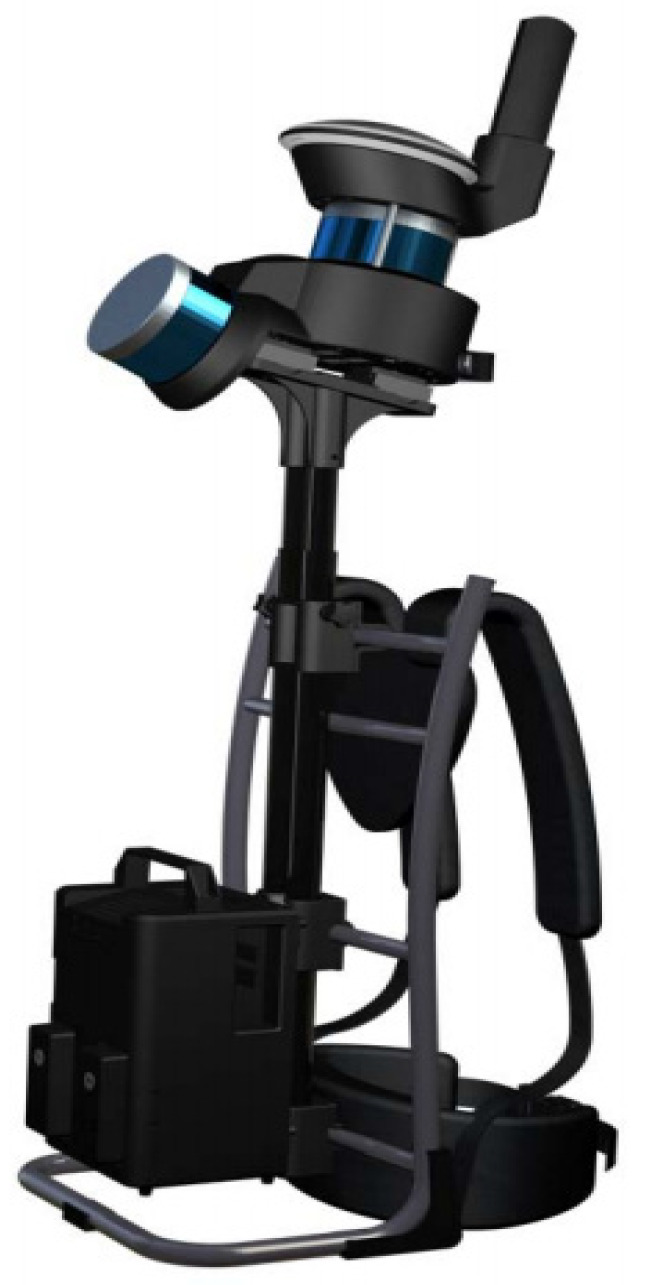
Backpack-Mounted LiDAR.

**Figure 15 sensors-24-05723-f015:**
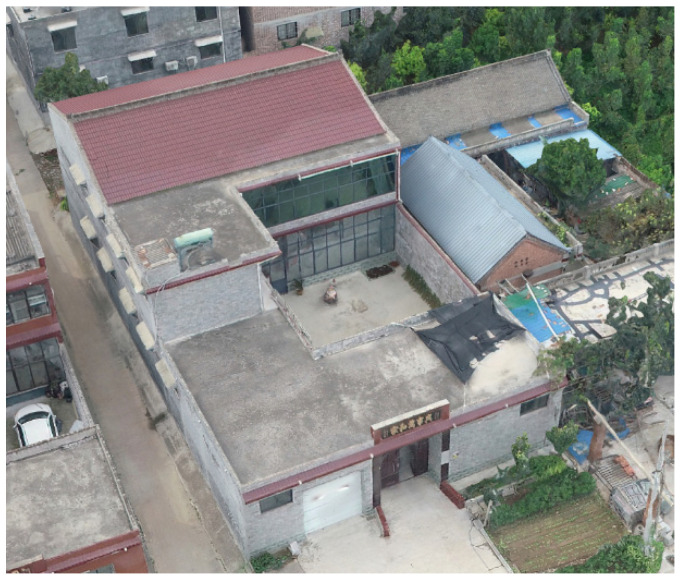
Field photo of the study village.

**Figure 16 sensors-24-05723-f016:**
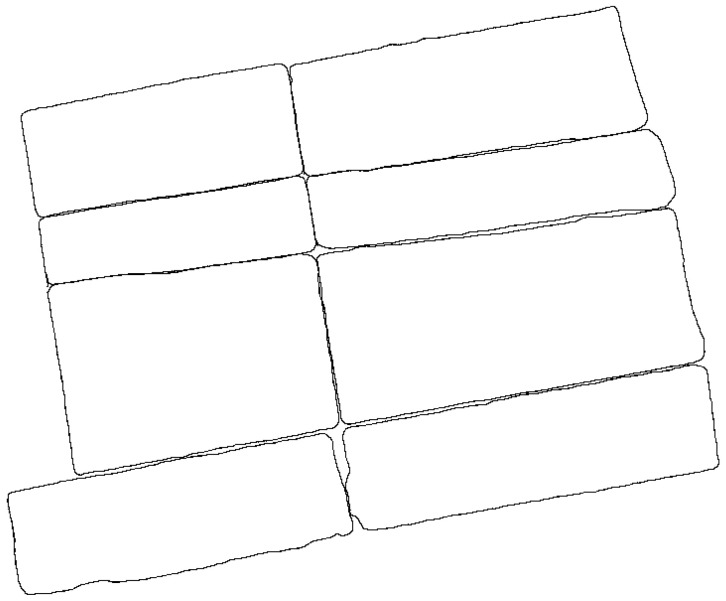
Data collection walking route.

**Figure 17 sensors-24-05723-f017:**
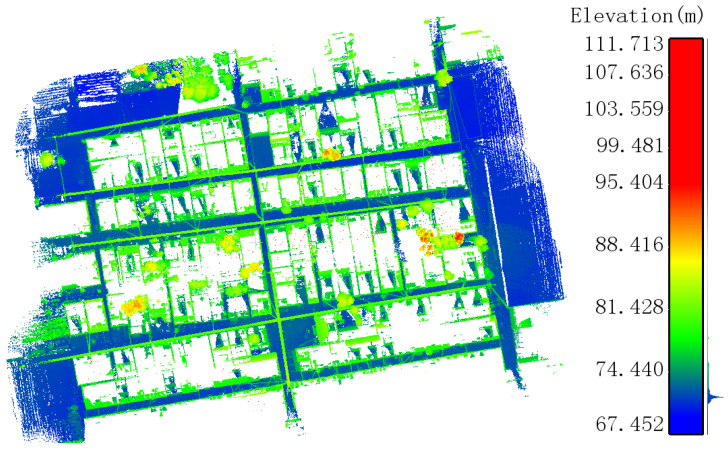
Outdoor point cloud data of rural houses.

**Figure 18 sensors-24-05723-f018:**
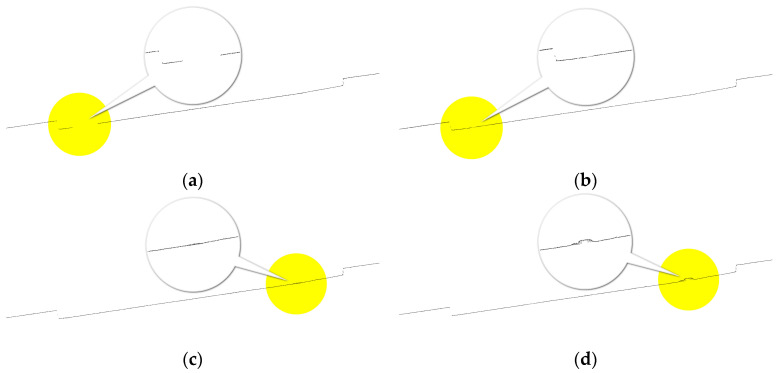
Walls at different height clipping ranges. (**a**) Wall clipping range of 1.3~1.5 m (**b**) Wall clipping range of 1.5~1.7 m (**c**) Wall clipping range of 1.7~1.9 m (**d**) Wall clipping range of 1.9~2.1 m.

**Figure 19 sensors-24-05723-f019:**
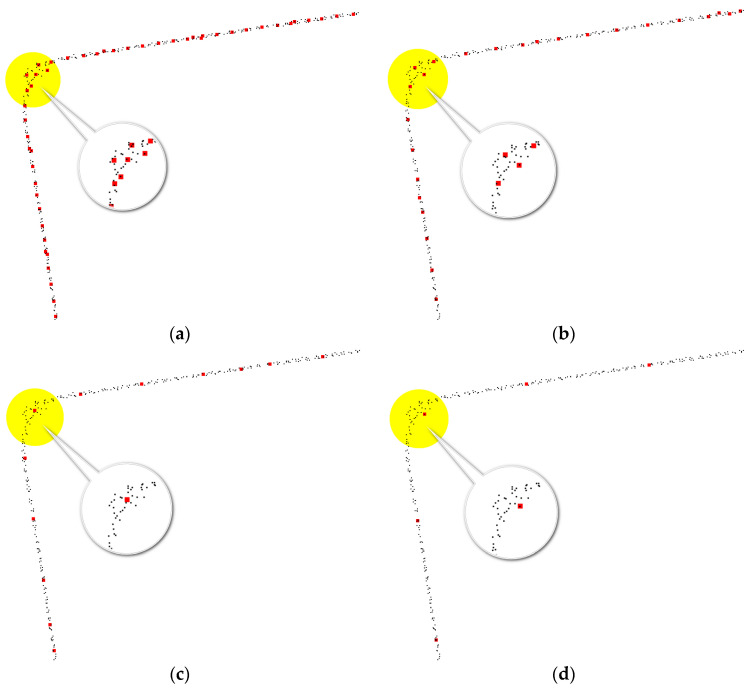
Walls at different grid sizes. (**a**) Grid size of 0.05 m (**b**) Grid size of 0.10 m (**c**) Grid size of 0.20 m (**d**) Grid size of 0.40 m.

**Figure 20 sensors-24-05723-f020:**
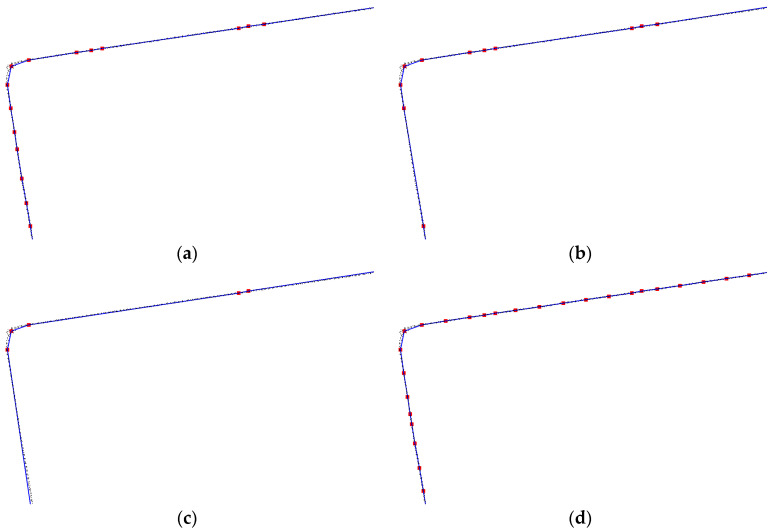
Floor plans at different angle thresholds. (**a**) Angle threshold of 179.5° (**b**) Angle threshold of 179.0° (**c**) Angle threshold of 178.0° (**d**) Initial floor plan.

**Figure 21 sensors-24-05723-f021:**
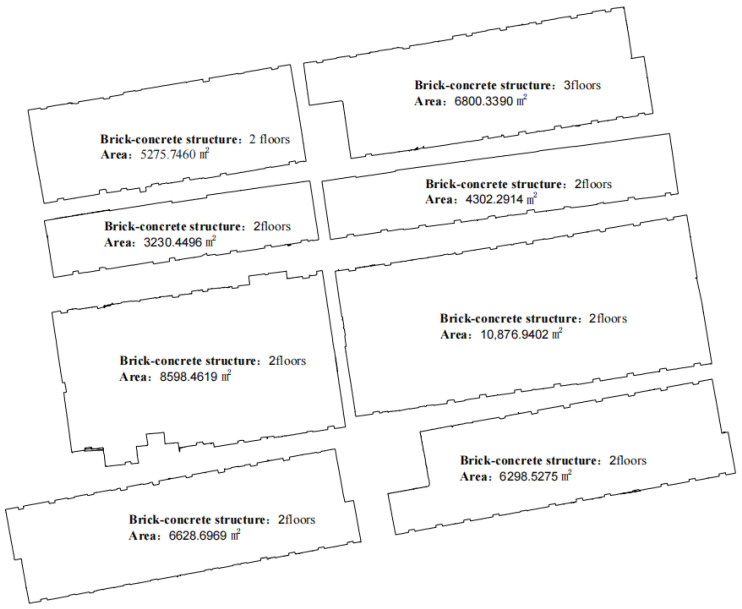
Outdoor floor plans of rural houses.

**Figure 22 sensors-24-05723-f022:**
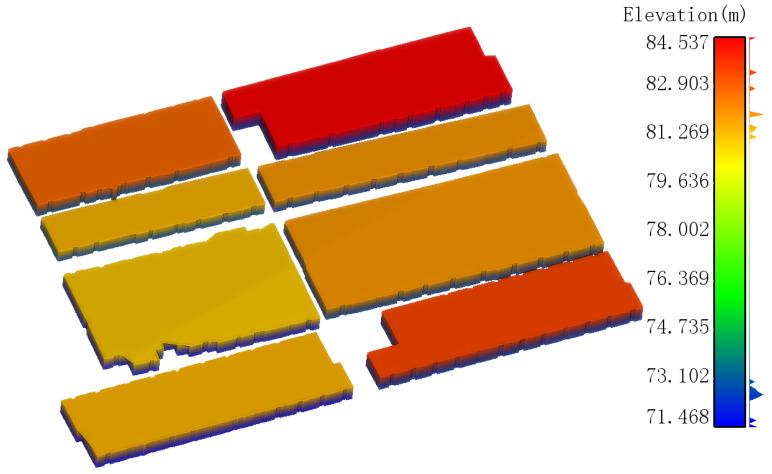
Outdoor 3D models of rural houses.

**Figure 23 sensors-24-05723-f023:**
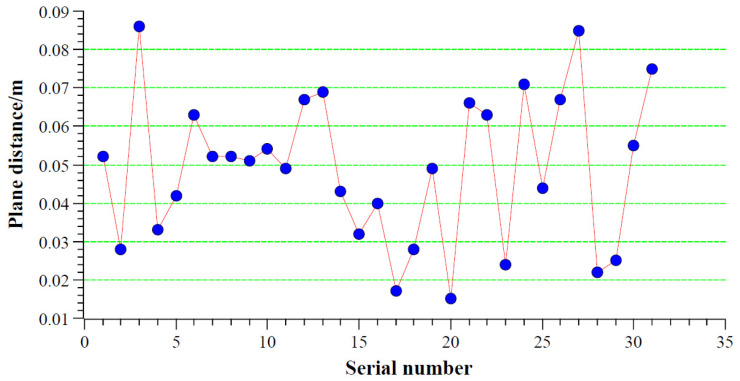
Planar distance between RTK coordinates and corresponding floor plan positions.

**Figure 24 sensors-24-05723-f024:**
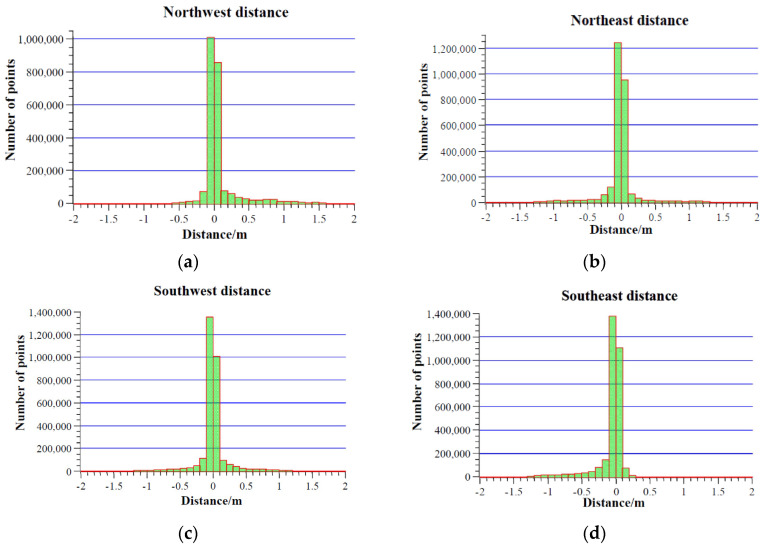
Distance from outdoor wall point cloud of rural houses to outdoor 3D models. (**a**) Northwest direction (**b**) Northeast direction (**c**) Southwest direction (**d**) Southeast direction.

**Table 1 sensors-24-05723-t001:** Parameters of backpack laser SLAM system.

**System Parameters**	Dimensions	980 × 260 × 310 mm	Positioning Principle	Laser SLAM+RTK
	LiDAR	16 Lines × 2	Camera/plus	Panorama/Ladybug5+
	Applicable Environment	Indoor and Outdoor Walkable Scenarios	Operation Mode	Backpack/Lightweight
	Processor	Quad-Core, Eight-Thread	Acquisition Speed	<20 km/h
**Laser Parameters**	LiDAR Accuracy	±3 cm	Scanning Field of View	360° × 360°
	Measurement Range	0.5~120 m	Scanning Frequency	600,000 points per second
**Data Output**	Relative Accuracy	≤3 cm	Absolute Accuracy	≤10 cm
	Point Cloud Format	Las, Pcd, Ply	Point Cloud Density	1~3 cm
	Point Cloud Thickness	<5 mm	Panorama Density	2~6 m

**Table 2 sensors-24-05723-t002:** Comparison of simplified floor plan and initial floor plan.

Type of Floor Plan	Angle Threshold	Number of Vertices	Area (m^2^)	Reduction Ratio (Compared to Initial Floor Plan)	
				Number of Vertices	Area
Simplified Floor Plan	179.5°	934	4302.2914	55.33%	0.001‰
	179.0°	523	4302.0247	74.99%	0.064‰
	178.0°	326	4301.9962	84.41%	0.071‰
Initial Floor Plan	None	2091	4302.2980	0.00%	0.000‰

**Table 3 sensors-24-05723-t003:** Accuracy analysis of RTK coordinates corresponding to floor plan positions.

ID	RTK Coordinates		Floor Plan Coordinates		Coordinate Difference		Planar Distance (m)	PDOP	HRMS	VRMS
	North Coordinate (m)	East Coordinate (m)	North Coordinate (m)	East Coordinate (m)	North Coordinate (m)	East Coordinate (m)				
1	138.957	−175.440	139.006	−175.423	0.049	0.017	0.052	1.8	0.014	0.023
2	164.145	−179.728	164.135	−179.702	−0.010	0.026	0.028	1.9	0.013	0.021
3	171.847	−179.703	171.890	−179.628	0.043	0.075	0.086	1.6	0.018	0.027
4	214.377	−187.300	214.356	−187.275	−0.021	0.025	0.033	1.6	0.015	0.026
5	235.707	−68.154	235.679	−68.123	−0.028	0.031	0.042	2.2	0.016	0.028
6	191.697	−60.736	191.709	−60.798	0.012	−0.062	0.063	2.7	0.02	0.034
7	183.036	−58.998	182.991	−59.024	−0.045	−0.026	0.052	1.9	0.024	0.032
8	156.058	−54.905	156.083	−54.951	0.025	−0.046	0.052	1.3	0.019	0.028
9	142.260	−53.270	142.212	−53.286	−0.048	−0.016	0.051	1.3	0.021	0.032
10	70.699	−42.857	70.717	−42.908	0.018	−0.051	0.054	1.4	0.017	0.03
11	60.952	−47.524	60.936	−47.570	−0.016	−0.046	0.049	1.7	0.018	0.034
12	18.770	−35.768	18.834	−35.788	0.064	−0.020	0.067	1.7	0.019	0.035
13	−8.839	−186.833	−8.779	−186.867	0.060	−0.034	0.069	1.5	0.014	0.025
14	10.856	−193.553	10.857	−193.596	0.001	−0.043	0.043	2.4	0.017	0.021
15	33.756	−197.546	33.730	−197.528	−0.026	0.018	0.032	1.4	0.014	0.028
16	59.482	−167.310	59.502	−167.345	0.020	−0.035	0.040	1.7	0.015	0.023
17	21.680	−19.942	21.693	−19.931	0.013	0.011	0.017	1.4	0.016	0.028
18	68.650	−8.937	68.651	−8.909	0.001	0.028	0.028	1.4	0.017	0.021
19	75.405	−38.107	75.450	−38.088	0.045	0.019	0.049	1.4	0.014	0.022
20	141.834	−47.579	141.845	−47.569	0.011	0.010	0.015	1.3	0.018	0.029
21	156.088	−49.684	156.085	−49.618	−0.003	0.066	0.066	1.2	0.026	0.038
22	182.702	−53.625	182.660	−53.672	−0.042	−0.047	0.063	2	0.024	0.036
23	217.149	−59.515	217.163	−59.495	0.014	0.020	0.024	1.6	0.018	0.029
24	236.507	−61.909	236.440	−61.887	−0.067	0.022	0.071	1.8	0.018	0.026
25	261.873	83.875	261.833	83.893	−0.040	0.018	0.044	1.4	0.016	0.196
26	213.442	97.105	213.507	97.120	0.065	0.015	0.067	1.6	0.018	0.025
27	204.477	104.484	204.442	104.406	−0.035	−0.078	0.085	1.9	0.019	0.029
28	167.756	112.232	167.734	112.228	−0.022	−0.004	0.022	1.8	0.022	0.031
29	99.910	123.789	99.912	123.764	0.002	−0.025	0.025	1.5	0.017	0.028
30	93.095	123.961	93.040	123.958	−0.055	−0.003	0.055	1.9	0.016	0.025
31	50.152	134.718	50.120	134.650	−0.032	−0.068	0.075	1.7	0.013	0.024
		Average Planar Distance (m)			0.049		Mean Error (m)		0.019	

**Table 4 sensors-24-05723-t004:** Precision of plane position of ground object points.

Regional Distribution	Scale	Point Location Mean Error	Mean Error of Distance between Adjacent Feature Points
Urban, Plain, Hilly Area, Industrial Building Area	1:500	±0.30	±0.20
	1:1000	±0.60	±0.40
	1:2000	±1.20	±0.80

## Data Availability

The datasets used and/or analysed during the current study available from the corresponding author on reasonable request.

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
