# Peer review of "An Algorithm for Generating Outdoor Floor Plans and 3D Models of Rural Houses Based on Backpack LiDAR"

_sensors, 2024, doi:10.3390/s24175723_

Round 1

Reviewer 1 Report

Comments and Suggestions for Authors

1. what is authoritative and new about this solution?

2. keywords and other places - LiDAR than lidar

3. fig 1b - please add some description to these figure. It is only the histogram of the point cloud from MLS backpack system? in my opinion it is rather from ALS/UAV LiDAR point cloud. When you are walking/driving in a narrow street with the MLS set then the roofs are not visible, and thus the cloud on the roof will not be as numerous as Ground. Even more so with 2 or 3 story buildings - see additional reference.

4. line 104 - Revit, ArchiCAD or Rhino. SketchUp is for very patience users

5. line 105 - manually drawing the models - rather manually creating/preparing the models. drawing for 2D floor plans or cross sections

6. line 118 - what accuracy parameters characterized the RTK measurement? RMS horizontal, RMS vertical? where were the RTK fixtures taken from? what are the accuracy characteristics of points 1-31 from table 2? RMSxy, RMSh, PDOP, HDOP

7. line 119 - accuracy rewuirement for scale 1:1000 - it is better to support this with values from standards or laws.

8. line 132 - longitudinal stretch? maybe rather stretching the outline of the walls upwards?, or along the H axis in the (X, Y, H) coordinates system

9. line 201 - is it only the 180 deg condition that is checked? what if the points are arranged in the shape of a “tight” U? what is the condition for stopping simplification? what about objects that are not similar to a rectangle?

10. line 219 - where does “h” come from in equation (5). Hh is unlikely to come out an integer - what happens to Hh when the value has decimal places?

11. line 228 - OK, I understand that geometric attributes can be added from automation/calculation to the centroid, but where does the material information come from there?

12. line 268 - what if the outdoor plan on the ground and below the roof are different in shape?

13. line 287 - the approach presented will not reflect the actual shape of the roof. the solution will work for a flat roof, and as you can see in fig. 15 not all roofs are flat. a gabled roof will be “skewed”, while a 4 pitch roof will be completely flattened.

14. line 328 - not “we”, rather Authors, or impersonal

15. line 366 - are exterior 3D models needed for anything? lighting was mentioned, so windows would be needed in such a model because sunlight access to windows is important, not to walls. what about windows inside the quarter?

16. what was the density of the acquired LiDAR point cloud? what was the accuracy of the LiDAR project assembly/alignment? was the cloud georeferenced? if so, how?

17. in my opinion GNSS RTK is not an accurate enough measurement technology to check MLS backpack clouds. total station would be better

18. fig. 24 - in my opinion the histograms “from a distance” to a little too little to be sure that the points are between -0.1 and +0.1 - from this it follows that the accuracy of the modeling is 0.2m? is this a sufficient value? in my opinion this is a little too large a range

19. line 401/402 - “by backpack LiDAR achieve an accuracy level that meets the 1:1000 topographic map standard” - please use the numbers

20. The literature list is a bit poor - interesting information can be found in:

- TLS and MLS density and no roofs (fig.10) - https://doi.org/10.5194/isprs-archives-XLVIII-1-W3-2023-205-2023

- MLS review - https://doi.org/10.1080/19475705.2021.1964617

Author Response

 We are particularly grateful to the reviewer for his/her detailed suggestions! According to the comments, we have tried our best to improve the previous manuscript. An item-by-item response to each constructive comment follows.

Comments 1.1

What is authoritative and new about this solution?

Response: We greatly appreciate the your question. Using 3D laser scanning for building data acquisition is a common method [1]. As the surveying industry advances towards informatization and intelligence, LiDAR technology, with its advantages of non-contact measurement, high speed, and high precision, plays a significant role in building surveys. Therefore, employing 3D laser scanning in rural building modeling is well-founded.

The innovation of this study lies in the use of backpack lidar, specifically targeting rural buildings, which brings a certain uniqueness. In the experiment, backpack lidar was used for data collection on rural buildings. Obtaining foundational data in rural settings requires selecting appropriate measurement instruments. Ground-based stationary lidar has a wide range of use and high measurement accuracy; however, due to the large area and varied layout of rural residences, multiple setups are required for data collection, making post-processing stitching difficult. Vehicle-mounted 3D lidar can quickly obtain data on both sides of roads, but rural roads are often narrow and enclosed, complicating route planning. Airborne lidar can rapidly acquire large-scale point cloud data, but due to laser scanning angle limitations, the point clouds of walls are very sparse. We opted for backpack lidar because of its compact size, suitable for data collection in walkable areas. Additionally, it can integrally capture both indoor and outdoor data of rural residences, avoiding the trouble of manually stitching indoor and outdoor data. Therefore, using backpack lidar for the digital reconstruction of rural residences is of great significance.

Comments 1.2

keywords and other places - LiDAR than lidar

Response: We greatly appreciate the your question. We have updated all instances of "LiDAR" to "lidar" in the manuscript.

Comments 1.3

fig 1b - please add some description to these figure. It is only the histogram of the point cloud from MLS backpack system? in my opinion it is rather from ALS/UAV LiDAR point cloud. When you are walking/driving in a narrow street with the MLS set then the roofs are not visible, and thus the cloud on the roof will not be as numerous as Ground. Even more so with 2 or 3 story buildings - see additional reference.

Response: We greatly appreciate the your question. Apologies for the lack of clarity in the previous statement. Figure 1b shows a histogram of building elevation information generated from indoor point cloud data collected by backpack lidar. The indoor data includes point clouds of the ground and roof, better reflecting the uniformity of point cloud distribution. This point has been clarified in the revised manuscript.

The revised manuscript is as follows: Statistical Histogram Based on Building Elevation Information: Figure 1b shows a histogram of building elevation information generated from indoor point cloud data collected by backpack lidar. The indoor data includes point clouds of the ground and roof, better reflecting the uniformity of point cloud distribution.

Comments 1.4

line 104 - Revit, ArchiCAD or Rhino. SketchUp is for very patience users

Response: We greatly appreciate the your suggestion. We mentioned SketchUp because we used it in previous research. We have modified the original manuscript as per your suggestion, as follows:

The revised manuscript is as follows: Currently, the most commonly used method for generating 3D models of buildings involves first using lidar equipment to collect point cloud data of the buildings, then importing this data into existing 3D modeling software (such as Revit, ArchiCAD or Rhino), and finally manually drawing the models.

Comments 1.5

line 105 - manually drawing the models - rather manually creating/preparing the models. drawing for 2D floor plans or cross sections

Response: We greatly appreciate the your suggestion. We have changed the expression of the original manuscript, which is as follows:

The revised manuscript is as follows: Currently, the most commonly used method for generating 3D models of buildings involves first using lidar equipment to collect point cloud data of the buildings, then importing this data into existing 3D modeling software (such as Revit, ArchiCAD or Rhino), and finally manually creating the models.

Comments 1.6

line 118 - what accuracy parameters characterized the RTK measurement? RMS horizontal, RMS vertical? where were the RTK fixtures taken from? what are the accuracy characteristics of points 1-31 from table 2? RMSxy, RMSh, PDOP, HDOP

Response: We greatly appreciate the your question. We have listed the precision parameters and characteristics of RTK measurements in the table below, with horizontal accuracy as HRMS and vertical accuracy as VRMS. The RTK device was purchased from Stonex, and the exported data does not include HDOP information.

Table  Precision parameters of RTK measurement

ID

North coordinates

East coordinates

Origin number

PDOP

HRMS

VRMS

1

138.957

-175.440

Pt1

1.8

0.014

0.023

2

164.145

-179.728

Pt2

1.9

0.013

0.021

3

171.847

-179.703

Pt3

1.6

0.018

0.027

4

214.377

-187.300

Pt4

1.6

0.015

0.026

5

235.707

-68.154

Pt5

2.2

0.016

0.028

6

191.697

-60.736

Pt38

2.7

0.02

0.034

7

183.036

-58.998

Pt37

1.9

0.024

0.032

8

156.058

-54.905

Pt34

1.3

0.019

0.028

9

142.260

-53.270

Pt33

1.3

0.021

0.032

10

70.699

-42.857

Pt29

1.4

0.017

0.03

11

60.952

-47.524

Pt28

1.7

0.018

0.034

12

18.770

-35.768

Pt22

1.7

0.019

0.035

13

-8.839

-186.833

Pt23

1.5

0.014

0.025

14

10.856

-193.553

Pt25

2.4

0.017

0.021

15

33.756

-197.546

Pt26

1.4

0.014

0.028

16

59.482

-167.310

Pt27

1.7

0.015

0.023

17

21.680

-19.942

Pt19

1.4

0.016

0.028

18

68.650

-8.937

Pt31

1.4

0.017

0.021

19

75.405

-38.107

Pt30

1.4

0.014

0.022

20

141.834

-47.579

Pt32

1.3

0.018

0.029

21

156.088

-49.684

Pt35

1.2

0.026

0.038

22

182.702

-53.625

Pt36

2

0.024

0.036

23

217.149

-59.515

Pt7

1.6

0.018

0.029

24

236.507

-61.909

Pt6

1.8

0.018

0.026

25

261.873

83.875

Pt13

1.4

0.016

0.196

26

213.442

97.105

Pt11

1.6

0.018

0.025

27

204.477

104.484

Pt12

1.9

0.019

0.029

28

167.756

112.232

Pt14

1.8

0.022

0.031

29

99.910

123.789

Pt15

1.5

0.017

0.028

30

93.095

123.961

Pt16

1.9

0.016

0.025

31

50.152

134.718

Pt18

1.7

0.013

0.024

Comments 1.7

line 119 - accuracy rewuirement for scale 1:1000 - it is better to support this with values from standards or laws.

Response: We greatly appreciate the your suggestion. Our proposed algorithm can meet the requirements for drawing 1:1000 scale rural building plans, as detailed in the table below.

Table  Precision of plane position of ground object points

Regional Distribution

Scale

Point Location Mean Error

Mean Error of Distance Between Adjacent Feature Points

UrbanPlainHilly AreaIndustrial Building Area

1:500

±0.30

±0.20

1:1000

±0.60

±0.40

1:2000

±1.20

±0.80

(https://openstd.samr.gov.cn/bzgk/gb/newGbInfo?hcno=D08E4189BD49B226725AFDA98B65E1BB)

Comments 1.8

line 132 - longitudinal stretch? maybe rather stretching the outline of the walls upwards?, or along the H axis in the (X, Y, H) coordinates system

Response: We greatly appreciate the your question. Longitudinal stretching is performed along the H-axis in the (X, Y, H) coordinate system. During stretching, we did not consider wall protrusions or recesses, such as eaves and air conditioner units, assuming the walls to be smooth planes.

Comments 1.9

line 201 - is it only the 180 deg condition that is checked? what if the points are arranged in the shape of a “tight” U? what is the condition for stopping simplification? what about objects that are not similar to a rectangle?

Response: We greatly appreciate the your question. In the part of simplifying the plan using the line segment angle threshold, simplifying line segments with angles close to 180° can improve the algorithm's efficiency. For U-shaped and other non-rectangular objects, due to their rich details, no simplification is done to better reflect their geometric characteristics. In future research, we will continue to explore more wall types and extract point clouds accordingly.

The revised manuscript is as follows: The condition for stopping the simplification is: generating new line segments clockwise or counterclockwise according to the topological relationship of the initial plan vertices until the initial point is recorded again, thus simplifying the redundant values between wall points.  (This point has been explained in the revised manuscript.)

Comments 1.10

line 219 - where does “h” come from in equation (5). Hh is unlikely to come out an integer - what happens to Hh when the value has decimal places?

Response: We greatly appreciate the your question. Apologies for the lack of clarity in the previous statement.  We have provided clarification in the revised manuscript.

The revised manuscript is as follows: The height of each floor of the specified rural residence is denoted by "h," with [ ] representing the floor function (rounding down).

Comments 1.11

line 228 - OK, I understand that geometric attributes can be added from automation/calculation to the centroid, but where does the material information come from there?

Response: We greatly appreciate the your question. Material information data is recorded during on-site collection and manually imported as attribute information before running the algorithm, after which it is automatically loaded into the floor plan.                                  

Comments 1.12

line 268 - what if the outdoor plan on the ground and below the roof are different in shape?

Response: We greatly appreciate the your question. Excluding the roof, the floor plans of rural buildings in the North China Plain are typically consistent between the ground level and below the roof, with only minor deviations in special cases.

Comments 1.13

line 287 - the approach presented will not reflect the actual shape of the roof. the solution will work for a flat roof, and as you can see in fig. 15 not all roofs are flat. a gabled roof will be “skewed”, while a 4 pitch roof will be completely flattened.

Response: We greatly appreciate the your suggestion. Your suggestions are very correct. The outdoor data of rural residences collected using backpack lidar in this study does not include roof data. For the roof part, we plan to use drone lidar in the future to generate roof data, which will then be stitched together to make the 3D model more accurate.

Comments 1.14

line 328 - not “we”, rather Authors, or impersonal

Response: We greatly appreciate the your question. We have made the necessary changes in the manuscript.

The revised manuscript is as follows: Subsequently, this study will optimize the parameters needed for the outdoor floor plan drawing algorithm to ensure the best possible results in terms of the appearance and accuracy of the generated plans.

Comments 1.15

line 366 - are exterior 3D models needed for anything? lighting was mentioned, so windows would be needed in such a model because sunlight access to windows is important, not to walls. what about windows inside the quarter?

Response: We greatly appreciate the your suggestion. Your suggestions are very constructive. We plan to classify and model objects on the walls (such as windows and doors) in the next experiments to make the 3D model more accurate.

Comments 1.16

what was the density of the acquired LiDAR point cloud? what was the accuracy of the LiDAR project assembly/alignment? was the cloud georeferenced? if so, how?

Response: We greatly appreciate the your question. After lidar point cloud processing and resampling, the point cloud density is 1-3 cm. The instrument's absolute accuracy is ≤10 cm, and relative accuracy is ≤3 cm. The point cloud has geographic references, using RTK-SLAM technology, with the coordinate system being CGCS2000, 3° zone. However, due to the confidentiality of the original experimental data, we can only present it using a local coordinate system.

Comments 1.17

in my opinion GNSS RTK is not an accurate enough measurement technology to check MLS backpack clouds. total station would be better.

Response: We greatly appreciate the your suggestion. When using a total station for single-point measurements, the field coordinates of objects are first measured and a sketch of spatial relationships between objects is drawn. This is then imported into software like CAD for manual topographic map drawing. This method is simple to operate, has low instrument costs, and high single-point measurement accuracy. However, it also has disadvantages such as high labor intensity, long operation periods, and low production efficiency. To address the issue of lacking total station measurement data for accuracy validation, considering that rural residential buildings generally have heights within 10 meters and village roads average 4 meters in width with clear visibility, the RTK measurement results in this study were obtained using a Qianxun CORS account under fixed solution conditions, with both horizontal and vertical accuracies within 2 cm. Additionally, the relative accuracy of the backpack lidar point cloud data is 3 cm, with absolute accuracy within 5 cm. Therefore, using RTK measurement results as a basis for accuracy validation is entirely reliable.

Comments 1.18

fig. 24 - in my opinion the histograms “from a distance” to a little too little to be sure that the points are between -0.1 and +0.1 - from this it follows that the accuracy of the modeling is 0.2m? is this a sufficient value? in my opinion this is a little too large a range

Response: We greatly appreciate the your question. The model was drawn based on the floor plan generated by the algorithm, with a horizontal accuracy of within 0.1 meters for the floor plan, resulting in the horizontal deviation of the generated 3D model walls also fluctuating around this range.

Comments 1.19

line 401/402 - “by backpack LiDAR achieve an accuracy level that meets the 1:1000 topographic map standard” - please use the numbers

Response: We greatly appreciate the your suggestion. Compared to the house corner coordinates measured by RTK, according to relevant standards, the plan generated from the outdoor point cloud of rural residences collected using backpack LiDAR can meet the 1:1000 topographic map plan accuracy requirement of 10 cm.

Comments 1.20

The literature list is a bit poor - interesting information can be found in:

- TLS and MLS density and no roofs (fig.10) - https://doi.org/10.5194/isprs-archives-XLVIII-1-W3-2023-205-2023      - edge effect - https://gll.urk.edu.pl/zasoby/74/GLL-3-1-2017.pdf

- MLS review - https://doi.org/10.1080/19475705.2021.1964617

Response: We greatly appreciate the your suggestion. We have added the above references to the manuscript.

References

  1. Li, X.; Qiu, F.; Shi, F.; Tang, Y. W., A Recursive Hull and Signal-Based Building Footprint Generation from Airborne LiDAR Data. REMOTE SENSING 2022, 14, (22).

Reviewer 2 Report

Comments and Suggestions for Authors

Based on the point cloud data collected by backpack LiDAR, this paper proposes an outdoor plan drawing algorithm based on the topological relationship between slice and grid wall point clouds. By comparing with the house corner coordinates measured by RTK, it is verified that the plan drawn by this algorithm can meet the accuracy requirements of 1:1000 topographic map.

Questions to consider before being accepted:

1. As can be seen from the title and the second half of the introduction, this paper actually proposes a novel algorithm, but it is not difficult to see that it is based on backpack LiDAR. Therefore, I think it is appropriate to add the specific parameters and structure diagram of the backpack liDAR used in this paper in the appropriate section. This is helpful to better let the reader know which backpack liDAR or which parameter liDAR is suitable for the proposed algorithm. Figure 14 is obviously not enough; It's not representative.

2. The selection of research scope focuses on the typical self-built houses in North China. Is it a representative type of structure? Is it necessary to study this type of building with liDAR?

3. Table 2 should highlight the results rather than simply list them.

4. The conclusion seems to be that there are only three steps of algorithm implementation, so it is suggested to condense it again. Based on the point cloud data collected by backpack LiDAR, this paper proposes an outdoor plan drawing algorithm based on the topological relationship between slice and grid wall point clouds. By comparing with the house corner coordinates measured by RTK, it is verified that the plan drawn by this algorithm can meet the accuracy requirements of 1:1000 topographic map.

Comments on the Quality of English Language

minor editing

Author Response

We are particularly grateful to the reviewer for his/her detailed suggestions! According to the comments, we have tried our best to improve the previous manuscript. An item-by-item response to each constructive comment follows.

Comments 2.1

As can be seen from the title and the second half of the introduction, this paper actually proposes a novel algorithm, but it is not difficult to see that it is based on backpack LiDAR. Therefore, I think it is appropriate to add the specific parameters and structure diagram of the backpack liDAR used in this paper in the appropriate section. This is helpful to better let the reader know which backpack liDAR or which parameter liDAR is suitable for the proposed algorithm. Figure 14 is obviously not enough; It's not representative.

Response:We greatly appreciate the your suggestion. We have added the specific parameters of the backpack liDAR in the revised manuscript in tabular form(zxnav.com/?products_13/),as shown below:

Table 1 Parameters of backpack laser SLAM system

System Parameters

Dimensions

980×260×310mm

Positioning Principle

Laser SLAM+RTK

LiDAR

16 Lines × 2

Camera/plus

Panorama/Ladybug5+

Applicable Environment

Indoor and Outdoor Walkable Scenarios

Operation Mode

Backpack/Lightweight

Processor

Quad-Core, Eight-Thread

Acquisition Speed

<20km/h

Laser Parameters

Lidar Accuracy

±3cm

Scanning Field of View

360°×360°

Measurement Range

0.5~120m

Scanning Frequency

600,000 points per second

Data Output

Relative Accuracy

≤3cm*

Absolute Accuracy

≤10cm

Point Cloud Format

Las,Pcd,Ply

Point Cloud Density

1cm~3cm

Point Cloud Thickness

<5mm

Panorama Density

2m~6m

Response:

Comments 2.2

The selection of research scope focuses on the typical self-built houses in North China. Is it a representative type of structure? Is it necessary to study this type of building with liDAR?

Response: We greatly appreciate the your question. The selected study area comprises typical self-built houses in North China.  Rural residences generally cover large areas, with most having varied layouts, narrow rural roads, and densely packed houses, which can be considered typical structures.  

Using LiDAR to study such buildings is necessary due to its advantages of non-contact measurement, high speed, and high accuracy, which are significant in building surveys.  We chose backpack LiDAR because of its compact size, making it suitable for data collection in walkable areas and capable of integrated indoor and outdoor data collection for rural residences, thus avoiding the trouble of manually stitching indoor and outdoor data.  Therefore, using backpack LiDAR as the primary data source to study the digital reconstruction of rural residences has significant research value.

Comments 2.3

Table 2 should highlight the results rather than simply list them.

Response: We greatly appreciate the your suggestion. Table 2 lists many results because the rural survey area is large and has numerous observation points.  To more clearly reflect the precision differences at various locations, we listed a larger number of corresponding points.These 31 selected points are representative in the real terrain, such as corner points and turning points.  We have provided a detailed analysis of Table 2 in the manuscript.

Comments 2.4

The conclusion seems to be that there are only three steps of algorithm implementation, so it is suggested to condense it again. Based on the point cloud data collected by backpack LiDAR, this paper proposes an outdoor plan drawing algorithm based on the topological relationship between slice and grid wall point clouds. By comparing with the house corner coordinates measured by RTK, it is verified that the plan drawn by this algorithm can meet the accuracy requirements of 1:1000 topographic map.

Response: We greatly appreciate the your suggestion. We have made changes in the manuscript as you suggested.

The revised manuscript is as follows: Based on the point cloud data collected by backpack lidar, this paper proposes an outdoor plan drawing algorithm based on the topological relationship between slice and grid wall point clouds. By comparing with the house corner coordinates measured by RTK, it is verified that the plan drawn by this algorithm can meet the accuracy requirements of 1:1000 topographic map.

Round 2

Reviewer 1 Report

Comments and Suggestions for Authors

Comments 1.1 - Your answer describes the advantages of backpack, but in this manuscript there are no indoor data. What is new? Scanning unit (backpack or GEO-VISIOM) - no. Preocessing -this is the SLAM solution from GEO-VISIOM I think. Products generate ? maybe.  Point it clearly in the manuscript. Have you developed software that generates products? Is it available somewhere for testing on other data?

A. LiDAR is correct - lidar incorrect - please change in the whole manuscript

Comments 1.3 - please add information about "indooring" of the point cloud to the description of the figure in line ""Figure 1. Method for extracting building wall point clouds

Comments 1.6 - please add information about PDOP, HRMS and VRMS to the table 3 in t the manuscript

Comments 1.7 - please add this table to the manuscript

Comments 1.11 - please add one sentence about this into the manuscript

Comments 1.13 - please add this information as "future works" in the Conclusion section

Comments 1.16 - please add this information (density, accuracy) into the manuscript

Comments 1.17 - "  high labor intensity, long operation periods, and low production efficiency" yes, it is true, but now we are on the experimental/research stage. so, in my opinion it is the best time to validate GEO-VISIOM point cloud. later, production will probably use RTK because it's faster, but we should investigate and report the accuracies that come out of a comparison of backpack MLS, total station and GNSS RTK. maybe as "future works" in the Conclusion section? To show that you are aware of the weaknesses of checking LiDAR on GNSS RTK.

Comments 1.18 - " horizontal accuracy of within 0.1 meters for the floor plan, resulting in the horizontal deviation of the generated 3D model walls also fluctuating around this range." do you have any proofs for this numbers?

Author Response

We are particularly grateful to the reviewer for his/her detailed suggestions! According to the comments, we have tried our best to improve the previous manuscript. An item-by-item response to each constructive comment follows.

Comments 1.1

Comments 1.1 - Your answer describes the advantages of backpack, but in this manuscript there are no indoor data. What is new? Scanning unit (backpack or GEO-VISIOM) - no. Preocessing -this is the SLAM solution from GEO-VISIOM I think. Products generate ? maybe.  Point it clearly in the manuscript. Have you developed software that generates products? Is it available somewhere for testing on other data?

Response: We greatly appreciate the your question. The backpack system is not considered a prominent innovation in this study. The SLAM solution from GEO-VISION was introduced primarily to diversify the data collection methods and does not represent a significant innovation. The novelty of this work lies in the two algorithms for generating outdoor floor plans and 3D models, an area with relatively few studies. Using data collected with the backpack system, we have validated the feasibility of these algorithms through empirical data, demonstrating their effectiveness in generating accurate floor plans and 3D models. These algorithms can be applied to other data testing scenarios. We have summarized these two algorithms as the key innovations in the conclusion section of the paper.

Comments 1.2

LiDAR is correct - lidar incorrect - please change in the whole manuscript

Response: We greatly appreciate the your question. We have updated all instances of " lidar " to "LiDAR" in the manuscript.

Comments 1.3

Comments 1.3 - please add information about "indooring" of the point cloud to the description of the figure in line ""Figure 1. Method for extracting building wall point clouds

Response: We greatly appreciate the your question. Apologies for the lack of clarity in the previous statement. Figure 1b shows a histogram of building elevation information generated from indoor point cloud data collected by backpack lidar. This point has been clarified in the revised manuscript.As follows:

The revised manuscript is as follows: (b) Histogram of indoor building elevation data

Comments 1.4

Comments 1.6 - please add information about PDOP, HRMS and VRMS to the table 3 in t the manuscript

Response: We greatly appreciate the your question. we added information about PDOP, HRMS and VRMS to the table 3.

Comments 1.5

Comments 1.7 - please add this table to the manuscript

Response: We greatly appreciate the your question. We added "Table 4. Precision of plane position of ground object points" to the manuscript.

Comments 1.6

Comments 1.11 - please add one sentence about this into the manuscript

Response: We greatly appreciate the your question. We added “Material information data is recorded during on-site collection and manually imported as attribute information before running the algorithm, after which it is automatically loaded into the floor plan” to the manuscript.   

Comments 1.7

Comments 1.13 - please add this information as "future works" in the Conclusion section

Response: We greatly appreciate the your question. We added "future works" in the Conclusion section.

Comments 1.8

Comments 1.16 - please add this information (density, accuracy) into the manuscript

Response: We greatly appreciate the your question. We added the information (density, accuracy) into the manuscript.

Comments 1.9

Comments 1.17 - "  high labor intensity, long operation periods, and low production efficiency" yes, it is true, but now we are on the experimental/research stage. so, in my opinion it is the best time to validate GEO-VISIOM point cloud. later, production will probably use RTK because it's faster, but we should investigate and report the accuracies that come out of a comparison of backpack MLS, total station and GNSS RTK. maybe as "future works" in the Conclusion section? To show that you are aware of the weaknesses of checking LiDAR on GNSS RTK.

Response: We greatly appreciate the your question. We added "future works" in the Conclusion section.

Comments 1.10

Comments 1.18 - " horizontal accuracy of within 0.1 meters for the floor plan, resulting in the horizontal deviation of the generated 3D model walls also fluctuating around this range." do you have any proofs for this numbers?

Response: We greatly appreciate the your question. Figure 24 presents histograms of the horizontal deviation of 3D model walls from four directions: northwest, northeast, southwest, and southeast.  The accuracy is almost entirely within 0.1 meters, with only a few outliers falling outside this range.  We appreciate your feedback on this section and will conduct a more detailed discussion in our future work.

Round 3

Reviewer 1 Report

Comments and Suggestions for Authors

Thank you for making changes to the manuscript